# Corrosion Behavior of Al_x_(CrFeNi)_1−x_ HEA under Simulated PWR Primary Water

**DOI:** 10.3390/ma15144975

**Published:** 2022-07-17

**Authors:** Dongwei Luo, Zhaoming Yang, Hengming Yang, Qingchun Chen, Yuan Wang, Nan Qiu

**Affiliations:** Key Laboratory of Radiation Physics and Technology of Ministry of Education, Institute of Nuclear Science and Technology, Sichuan University, Chengdu 610064, China; luodongwei0303@163.com (D.L.); yangzhaoming0721@163.com (Z.Y.); hengmingyang@163.com (H.Y.); qcchen2022@163.com (Q.C.)

**Keywords:** HEA, ATF cladding, corrosion resistance

## Abstract

High-entropy alloys (HEAs) have great potential as accident-tolerant fuel (ATF) cladding. Aluminum-forming duplex (BCC and FCC) stainless-steel (ADSS) is a candidate for ATF cladding, but the multiphase composition is detrimental to its corrosion resistance. In this paper, two single-phase HEAs were prepared by adjusting the content of each element in the ADSS alloy. The two HEAs were designed as Al_0.05_(CrFeNi)_0.95_(FCC) and Al_0.25_(FeCrNi)_0.75_(BCC). Their corrosion behavior under simulated pressurized water reactor (PWR) primary water was investigated. The corrosion products and corrosion mechanisms of these two HEAs were explored. The results show that the corrosion resistance of HEA alloys containing FCC is better than that of BCC and ADSS alloys. At the same time, the reason why the BCC structure composed of these four elements is not resistant to corrosion is revealed.

## 1. Introduction

Since the Fukushima disaster, zirconium alloy has revealed its major drawback as a nuclear fuel coating [1]. Subsequently, the concept of accident-tolerant fuel (ATF) cladding [2] was proposed, aiming to increase the tolerance time of the cladding under accident without degrading the performance of the fuel during normal operation. ATF claddings that have received more attention so far include metals (FeCrAl [3], Cr [4], Mo [5]) and ceramic coatings (SiC composite coating [6], TiAlN [7], etc.). However, these materials still face huge challenges in practical application. For example, Si in simulated pressurized water reactor (PWR) primary water produces H_2_SiO_3_ and dissolves [8]. Cr coating appears to peel and crack in high-temperature steam oxidation [4]. FeCrAl coating reacts with zirconium alloys at high temperatures [3], etc. Therefore, in the face of the severe service conditions of the reactor, it is still necessary to develop a new type of ATF cladding with stronger comprehensive performance. At the same time, based on the particularity of high-temperature water vapor oxidation, Al or Si almost must be considered in the design of ATF cladding materials [9].

In recent years, H. Kim et al. have prepared a set of alumina-forming duplex stainless-steel (ADSS) alloys [10] with a nominal composition range of Fe–(18–21)Ni–(16–21)Cr–(5–6)Al. Benefiting from the oxidation resistance of Al and Cr, this series of alloys have excellent performance in simulated PWR primary water corrosion and high-temperature steam oxidation. C. Kim et al. evaluated its mechanical properties and thin-walled tube fabrication methods in more detail [11], and believed that this series of alloys have the potential to become ATF cladding. However, the ADSS series alloys are dual-phase structures (including FCC and BCC) and contain a large amount of AlNi precipitates. Phase boundary and AlNi phase severely reduce the corrosion resistance of the alloy under the simulated PWR primary water [12]. High-entropy alloy [13] (HEA) is a new type of multi-principal single-phase alloy, which usually contains more than four principal elements, and the mole fraction of each principal element is between 5% and 35%. HEAs have excellent performance in mechanical properties [14], radiation resistance [15], oxidation resistance [16], corrosion resistance [17,18], etc., and are one of the candidates for new ATF cladding. For example, Zhao et al. used radio-frequency magnetron sputtering (RF) technology to prepare an AlTiCrNiTa coating with a thickness of 3 um on a zirconium alloy substrate to simulate the corrosion of PWR primary water for 45 days to produce NiCr_2_O_4_ on the surface, which can effectively protect the substrate from oxidation [19]. Tao et al. prepared AlCrFeCuNb_x_ bulk by arc melting. This high-entropy alloy has similar corrosion weight gain to the Zr-1Nb alloy, but the high-temperature steam oxidation rate is two orders of magnitude lower than that of zirconium alloy [20] Fe, Cr, Ni, and Al are the core elements of ADSS alloys. In this paper, two single-phase high-entropy alloys (BCC and FCC) were prepared by adjusting the content of four elements. The single-phase structure solves the key problem of poor corrosion resistance of ADSS alloy phase boundaries. The ATF cladding under normal working conditions is immersed in the PWR primary water for a long time, so the corrosion resistance of the cladding under this working condition will be an important guarantee for the safe operation of the reactor. Therefore, in this work, the corrosion behavior of these two HEAs under simulated PWR primary water was investigated. The corrosion products and corrosion mechanisms of these two HEAs are discussed in detail, and their corrosion resistance is evaluated.

## 2. Experiment

### 2.1. Preparation of Alloy Samples

The HEA was prepared by arc melting using Fe, Cr, Ni, and Al powders with a purity of more than 99.9%. Through experiments, two types of alloys, Al_0.05_(CrFeNi)_0.95_ (detected as FCC single phase) and Al_0.25_(FeCrNi)_0.75_ (detected as BCC single phase) were screened as research objects. The as-cast alloy ingot is a cylinder with a diameter of 80 mm and a height of 150 mm. The alloy ingot was processed into a cuboid with a size of 10 × 10 × 2 mm by wire cutting. Each surface was polished with #400, #800, #1500, #2000, and #3000 sandpaper in turn, and then polished for 20 min. The polished samples were ultrasonically cleaned with acetone, deionized water, and alcohol in turn, and then dried with nitrogen for further use.

### 2.2. Corrosion Test

A static autoclave was used to conduct the corrosion test. To simulate the solution environment under service conditions, an aqueous solution containing 1000 ppm of HBO_3_ and 3.5 ppm of LiOH (by weight) was prepared with pure water and injected into the reactor, with O content restricted to less than 10 ppm (by volume). The instrument’s working pressure was set to 11.3 MPa, the temperature was set to 320 °C, and the heating rate was set to 6 °C/min. After 20 days of corrosion, it was naturally cooled, and a group of samples was removed. The remaining samples were left in the autoclave, and the temperature was raised using the same parameters. Another bunch of samples from the autoclave was taken out after 20 days. The process was repeated 3 times to obtain water corrosion samples with exposure times of 20, 40, and 60 days. After removing the samples, they were washed with deionized water and dried with nitrogen for analysis.

### 2.3. Analysis Methods

Each sample was weighed three times with a balance that had an analytical accuracy of 0.1 mg, and the average weight was used to acquire the sample weight data. The microstructure and the elemental composition of the as-cast alloys were examined by scanning electron microscopy (ZEISS Gemini 300). An X-ray diffractometer (Rigaku Smartlab 9 kw) was used to examine the phase composition of the samples. After corrosion, an X-ray photoelectron spectrometer (Thermo Scientific K-Alpha) was used to detect the valence of the elements on the surface of the sample, which assists in determining the phase structure of the oxide film. A scanning electron microscope (TESCAN MIRA LMS) was used to characterize the microscopic morphology and composition of the corroded samples in detail.

## 3. Results and Discussion

### 3.1. Microstructure and Phase of HEA

According to the ratio of raw materials added, the two alloys with different Al contents were recorded as Al_0.25_(FeCrNi)_0.75_ and Al_0.05_(CrFeNi)_0.95_. There was no discernible difference between the EDS test findings of the two alloys and the ratio of raw materials added during casting. Figure 1c,d show that after grinding and polishing, the surfaces of the two alloys were smooth and flat, with few shallow scratches, ensuring that the corrosion results were less affected by the surface roughness [21]. After polishing the Al_0.25_(CrFeNi)_0.75_ alloy, obvious grain boundaries could be detected. The grain boundaries were about several hundred nm wide. The grain interior showed a regular grid-like distribution, which may be caused by spinodal decomposition. A similar situation has also occurred in AlCrFeNi eutectic alloys of similar composition [22], while Al_0.05_(CrFeNi)_0.95_ showed a smooth surface morphology.

The XRD results of Al_0.05_(CrFeNi)_0.95_ in Figure 1e correspond to the three strong peaks of the FCC structure, without other impurity peaks. At the same time, the EDS mapping results in Figure 2 show that the elements in the alloy are uniformly distributed, which is consistent with the multi-principal solid-solution characteristics of HEA. Therefore, it was judged that Al_0.05_(CrFeNi)_0.95_ alloy is a FCC single-phase HEA. The XRD results of Al_0.25_(CrFeNi)_0.75_ show that it is a single phase of BCC, and the mapping results show slight and regular element segregation. Therefore, it was judged that the Al_0.25_(CrFeNi)_0.75_ alloy is a BCC single-phase HEA, but there is a certain degree of spinodal decomposition. For the convenience of description, the BCC alloy is denoted as B_0_, and samples corroded in high-temperature water for 20, 40, and 60 days were denoted as B_20_, B_40_, and B_60_, respectively. Similarly, the FCC series samples are denoted as F_0_, F_20_, F_40_, and F_60_.

### 3.2. Phase Structure Analysis of Corrosion Samples

#### 3.2.1. XPS Analysis

To further identify the phase composition of oxides on F_60_ and B_60_, the XPS fine spectra of Al, Cr, Fe, Ni, and O on their surfaces were collected, and the results are shown in Figure 3 and Figure 4. The collected data were processed by Avantage, a single-peak fitting was used in s orbit, and a more persuasive bimodal fitting was used in 2p orbit. For 2p orbitals, the 2p_3/2_ energy level with a stronger signal was used as the basis of qualitative and semi-quantitative calibration. The 2p_1/2_ energy levels of Cr, Fe, and Ni are circled with black dotted lines. The strong peak near 68 eV was the 3p characteristic peak of nickel ion, while the peak at 80 eV was the corresponding satellite peak, which is circled by the blue dotted line in the picture. The Auger electron peak of Ni was near 705 eV and has been circled by green dotted lines. The two peaks near 861 eV were satellite peaks of nickel ion 2p_3/2_ energy levels, which are circled by orange dotted lines.

Figure 3 shows the XPS fine spectrum of Al, Cr, Fe, Ni, and O on the surface of the F_60_ sample. The clearly separated O element spectrum was fitted to two peaks, 529.07 eV corresponding to O^2−^ and 531.52 eV to OH^−^. The valence of Al’s binding energy was +3, and the characteristic peaks were only slightly higher than the curve noise, indicating that the Al content was very low. The Cr-broadened 2p_3/2_ peak was subdivided into two parts: 575.63 eV corresponding to the oxide of Cr^3+^ and 576.92 eV to the hydroxide of Cr^3+^. The fitted Fe spectrum revealed that Fe^2+^ and Fe^3+^ coexisted. After double-peak fitting, the fitted peak was lower than the experimental value at 2p_3/2_, but slightly higher than the experimental value at 2p_1/2_. It was speculated that the count was increased due to the coincidence of the Auger electron energy of nickel with the 2p_3/2_ energy level of Fe. The 853.95 and 855.96 eV of Ni were subdivided and corresponded to Ni^2+^ and Ni^3+^, respectively. The coexistence of a large number of O^2−^ and OH^−^ indicated that the larger half-width of the fitting peak of iron and nickel might be caused by the coexistence of hydroxides and oxides of the same valence [23]. This is because the oxides exposed to hot water undergo hydrolysis reactions [24]. Using the peak area and the sensitivity of the corresponding energy level, the relative content of metal elements was calibrated, while the count peaks with Al slightly higher than the curve noise were not involved in the comparison. The molar ratio of Cr, Fe, and Ni was calibrated to be 0.64:1.19:1, respectively. It shows that the surface of F_60_ is mainly composed of Cr, Fe, Ni, and O.

Figure 4 show the XPS fine spectrum of Al, Cr, Fe, Ni, and O on the surface of the B_60_ sample, with the broadened O1s peak being subdivided into 529.33 and 531.02 eV, corresponding to O^2−^ and OH^−^, respectively. The Al2p orbital fit peak corresponded to Al^3+^ at 73.19 eV, and the broadened energy fit peak indicated that more than one anion was paired with Al^3+^. The fitting results of the Cr spectrum show that in addition to a broadened Cr^3+^ peak, there was also a weak Cr^6+^ peak, and the high valence of chromium ions indicated that the surface of B_60_ may have a more abundant anionic environment than that of F_60_. The Ni2p spectrum exhibited excellent counting signals as well as broad fitting peaks. The large number of Auger electrons of Ni made the fine spectrum of Fe appear to be highly broadened between 700 and 710 eV, interfering with the analysis of Fe valence and content. The Fe content was calculated by counting the Fe 2p_1/2_ energy levels. In summary, the broadening of the fitted peaks of the same valence state and the coexistence of O^2−^ and OH^−^ at the same time indicate that the surface oxides of B_60_ were being hydrolyzed. The molar ratio of Al, Cr, Fe, and Ni was calibrated to be 19.0:3.1:1:13.7, which indicated that the metal elements on the surface of the B_60_ sample were mainly Al and Ni.

#### 3.2.2. XRD Analysis

Figure 5 shows the XRD patterns of the HEAs samples after corrosion. Figure 5a shows that the XRD results of B_20_, B_40_, and B_60_ samples were dominated by the BCC structure of the original alloy, and the intensity of oxidation peaks was low. With the increase of the etching time, there was no obvious enhancement of the oxide peak. The XRD results of F_20_, F_40_, and F_60_ samples also showed that the FCC structure of the original alloy was dominant, but the oxide peaks increased significantly with the increase of corrosion time. It is worth noting that the characteristic peaks of the BCC structure appeared in the XRD spectrum of F_60_, as shown by the orange dotted circle in the figure. The intensities of the oxidation peaks were weak compared to the intensities of the characteristic peaks of the base alloy, indicating that the surface oxides after corrosion of both HEAs were very thin.

To better analyze the phase composition of oxides, F_60_ and B_60_ with the longest corrosion time were characterized by GIXRD, and the results are shown in Figure 5b. Two grazing incident angles of 0.5° and 1° were selected for each sample. The orange dotted line circle of Figure 5a shows that F_60_ had two characteristic peaks (43.7° and 44.5°) near 44°. However, there is no sign of double peaks at the corresponding position of the GIXRD spectrum, and there is no obvious broadening of the peak shape. Combined with the cross-sectional distribution of elements in Figure 8, it is speculated that because the BCC structure in the F_60_ sample is near 2 μm from the surface, the grazing incidence cannot display its signal.

Figure 5b shows the GIXRD results of F_60_ and B_60_. The oxide peaks shown in the GIXRD pattern of F_60_ are basically consistent with the FeCr_2_O_4_ and NiCr_2_O_4_ peaks produced after etching in [16,18]. Considering that NiFe_2_O_4_, FeCr_2_O_4_, and NiCr_2_O_4_ have similar spatial configurations, all three are likely to be generated on the surface of F_60_. At the same time, the XPS results in Figure 3 show that the contents and valence states of Cr, Fe, and Ni on the surface of the sample all support the coexistence of the three oxides. The formation of these three oxides is predicted to be a competing process. Therefore, the oxide on the surface of F_60_ is abbreviated as (FeCrNi)_3_O_4_. There were few characteristic peaks in the GIXRD spectrum of the B_60_ sample. Considering all possible elements, only two oxides, Ni_1.94_O_3_H_0.81_ and triclinic Al_2_O_3_, can correspond to them. The XPS results in Figure 4 show that the surface of the B_60_ sample mainly contained Al and Ni elements, and the valence was consistent with the conclusion that the oxides are Ni_1.94_O_3_H_0.81_ and Al_2_O_3_.

### 3.3. Micromorphology after Corrosion

Figure 6 shows the SEM surface morphologies (10,000×) and the chemical compositions of the two alloys after 20 and 60 days of corrosion. Inside the solid orange frame is a partially magnified image (50,000×). Figure 6a,b shows that the longer the corrosion time of the F_0_ alloy, the denser the spinel particles on the surface, and a dense spinel protective layer formed on the surface after 60 days of corrosion. Spinel particle size distribution was between tens to hundreds of nm, with a large number of particles concentrated around 100 nm. The content of Al in the metal elements hardly changed before and after the corrosion of the F_0_ alloy, and no oxides of Al were found in the phase analysis. According to Gibbs free energy [25], Al_2_O_3_ is the most easily formed simple oxide in this system (Al, Cr, Fe, Ni). However, according to Tang [26] et al., the formation of the Al_2_O_3_ scale requires a minimum concentration of Al. This value is related to test conditions such as oxygen partial pressure and temperature. Therefore, it was judged that the 5% Al content in the F_0_ alloy may not reach this minimum concentration under the experimental conditions. As Liu et al. found [17], Al oxides were also not detected after corrosion of AlxCoCrFe HEA with an Al content of 6%. In the XPS results shown in Figure 3, no Al^3+^ ions were detected. To sum up, Al does not participate in the medium- and long-term corrosion process of the F_0_ alloy.

As shown in Figure 6c,d, compared with the B_0_ alloy, the EDS results of B_20_ and B_60_ show that the proportion of Ni in the metal element significantly increased. According to the phase analysis results, it was judged that Ni_1.94_O_3_H_0.81_ formed on the surface. The content of Al was significantly reduced, which is because the Al_2_O_3_ generated on the surface of the sample encounters the high-temperature water in the reaction kettle and undergoes a hydrolysis reaction, resulting in the formation of easily soluble AlOOH. This phenomenon is common in high-temperature water corrosion of Al-based alloys [24]. Comparing the corrosion time of the B_0_ alloy from 20 to 60 days, the proportion of Ni in the metal elements decreased from 35.0% to 31.2%. The proportion of Al in the metal elements decreased from 10.6% to 9.3%. The content of O decreased from 35.1% to 31.5%. The Al element continued to decrease with the increase of corrosion time, indicating that the hydrolysis reaction of Al_2_O_3_ was still taking place. From the observation of the surface morphologies of B_20_ and B_60_, it is obvious that the surface of B_60_ had less flake oxide attached and was obviously peeling off. Combined with the conclusion that the proportion of Ni and O was lower after 60 days of corrosion, it is speculated that the flake oxide that falls off the surface of the sample is Ni_1.94_O_3_H_0.81_. This is because the Al_2_O_3_ is hydrolyzed, which causes the oxide to be unstable and fall off. A similar situation has also occurred in the study by Sha et al. [24].

Figure 7 shows the surface SEM mapping of typical regions of F_60_ and B_60_ samples. The O content on the surface of the F_60_ sample was significantly lower than that of the B_60_ sample, indicating that the oxide film of B_60_ was thicker. The distribution of elements on the surface of F_60_ was basically uniform. In contrast, a large number of small-scale element segregation could be seen in the spectra of Fe, Cr, and Ni elements on the surface of the B_60_ sample, which corresponded to the grid-like structure of the B_0_ alloy in Figure 1c. Compared with the mapping results of the B_0_ alloy in Figure 2, the segregation degree of Cr, Fe, and Ni elements weakened after 60 days of corrosion, and Al was no longer distributed in the original grid. It shows that in the long-term corrosion process, each element in the B_0_ alloy has a certain degree of diffusion, and the Al element has the most serious diffusion.

The samples inlaid with epoxy resin were repeatedly sanded with #7000 sandpaper, ultrasonically cleaned, and dried with nitrogen. The cross-sectional micro-morphology of the sample was obtained, as shown in Figure 8. Obviously, the obtained section was not flat. Since the absolute count of the Line EDS will be affected by the surface topography, the normalization method was used in this paper to convert the count of each element in the Line EDS into the relative count rate. At the same time, some representative points were selected to measure the Point EDS. As an auxiliary description, the Point EDS data are organized in Table 1. According to the cross-sectional micro-morphology and elemental composition shown in Figure 8a,b, the cross-section was divided into alloy matrix, oxide layer, and epoxy resin coating layer from the bottom to the top. The thickness of the oxide layer of the F_60_ sample and the B_60_ sample was similar, about 4 um.

Figure 8c,d are the EDS results of Lines 1 and 2, respectively. According to the distribution of oxygen, the oxide layer was roughly divided into an outer oxygen-rich oxide film and an inner oxygen-deficient transition layer. The dividing line is shown as a pink dotted line. The oxygen content of F_60_ decreased rapidly at a distance of 700 nm from the surface, indicating that the dense (FeCrNi)_3_O_4_ layer formed on the surface of the F_60_ sample can effectively prevent oxygen intrusion and reduce the oxidation rate of the alloy, which is consistent with the conclusion of Terachi et al. [27]. Oxygen in B_60_ was enriched near the boundary between the oxide layer and the alloy matrix, and the accumulated oxygen content in the oxide layer was significantly higher than that of the F_60_ sample. Therefore, it is considered that the surface of the B_60_ sample was more severely oxidized.

The results of Line 1 show that the transition layer had a higher Al content than the base alloy. The content of each element in the matrix remained stable, so it was judged that the increase of Al content in the transition layer was caused by the diffusion of Cr, Fe, and Ni to the surface of the sample to form (FeCrNi)_3_O_4_. Point 1 shows a similar chemical composition to the B_0_ alloy, whereby the high Al region was mainly in the middle of the oxide layer, about 2 μm from the surface, which explains why the BCC phase could be observed in the XRD of F_60_ but no BCC characteristic peak was found in its GIXRD. During the corrosion process of the F_0_ alloy, there was no obvious loss of Al like that of the B_0_ corrosion process, which may be mainly based on the following two points: On the one hand, the content of Al was low, and the oxidation reaction was not strong enough. On the other hand, the diffusion of Al in FCC-structured F_0_ alloys seems to be difficult because even if high Al regions were generated in the transition layer, there was no diffusion to the surface along the concentration gradient.

The data for Line 2 show that Al was depleted at the surface due to AlOOH dissolution. In the BCC structure of this component, Al had the fastest migration rate to defect vacancies, followed by Cr, Fe, and Ni. This rule has been confirmed in the previous research of our group [28]. The Al on the surface was continuously dissolved to generate vacancies, resulting in the continuous diffusion of the elements inside the alloy. The diffusion rate of Al to the surface was the fastest, and Cr was the second, so the interface between the oxide layer and the transition layer exhibited Al depletion and Cr enrichment. On the other hand, the metal matrix showed a small area depleted of Al, Cr, and Fe, and enriched with Ni, as shown by the orange dotted circle in Figure 8d. This is the Ni retention phenomenon caused by the lower out-diffusion rate of Ni than that of Al, Cr, and Fe. The Al element diffused out the fastest and further increased the amount of Al_2_O_3_ formed by oxidation on the surface of the sample. According to the Gibbs free energy [29], Al_2_O_3_ was the most easily formed simple oxide in this system (Al, Cr, Fe, Ni). Subsequently, the hydrolysis of a large amount of Al_2_O_3_ led to the shedding of other oxide particles, leaving only a part of Ni_1.94_O_3_H_0.81_ with a sheet-like structure. The oxide was loose and porous, so more oxygen diffused into the interior of the transition layer.

Consider the quality of the oxide film. F_60_ showed a uniform and flat surface, and the oxide layer had no obvious cracking, peeling, or other defects. In the cross-section shown in Figure 8, no defects such as cracks and holes were found. Compared with B_60_, which was not resistant to oxidation, such as a large loss of Al and surface oxide shedding, it is obvious that the oxide film generated by F_60_ had better integrity and density. It can better protect the substrate. At the same time, compared with the corrosion results of ADSS alloys under similar conditions [30], the oxide film of the F_0_ alloy in this study after corrosion was thinner, the surface oxide particles were smaller and denser, so the protection was stronger, and the microscopic morphology of the corroded surface was more uniform and flatter.

### 3.4. Weight Gain after Corrosion

Figure 9 shows the weight gain of alloy samples in simulated PWR primary water as a function of corrosion time. Oxidation of the alloy will increase the weight of the sample, and the mass loss is due to the dissolution of metal elements or the removal of oxide particles [29,31]. The weight of the F_0_ alloy increased slowly with the corrosion time, which is the result of the slow growth of the oxide layer on the surface of the alloy. In contrast, the quality of the B_0_ alloy decreased with the corrosion time, because Al dissolves in high-temperature water to form AlOOH. At the same time, the dissolution of AlOOH causes other oxides to be unstable and fall off, which further aggravates the mass loss [32].

According to the quality change combined with the previous analysis, the F_0_ alloy protected the substrate from further oxidation by generating oxides on the surface. The B_0_ alloy achieved the purpose of protecting the substrate by maintaining a balance of oxidation and dissolution, which will cause the cladding to be continuously corroded and thinned, increasing the risk of tube wall cracking. From this point of view, dual-phase ADSS alloys may have certain potential risks.

## 4. Conclusions

In this study, two kinds of alloys, Al_0.05_(CrFeNi)_0.95_ and Al_0.25_(CrFeNi)_0.75_, were prepared by arc melting. After processing, the alloys were exposed to 320 °C, 11.3 MPa, high-temperature water containing HBO_3_ and LiOH for 20, 40, and 60 days, respectively. The weight change, phase structure, microscopic morphology, and composition of the oxidation product were analyzed in detail, and the following conclusions were obtained.

Al_0.05_(CrFeNi)_0.95_ is a FCC single-phase HEA (denoted as F_0_), and the oxidation product on the surface after corrosion is mainly (FeCrNi)_3_O_4_. The dense (FeCrNi)_3_O_4_ spinel particles formed a corrosion barrier, which greatly improved the corrosion resistance of the F_0_ alloy. Al_0.25_(CrFeNi)_0.75_ is a BCC single-phase HEA (referred to as B_0_), and the oxidation products on the surface after corrosion are mainly Ni_1.94_O_3_H_0.81_ and Al_2_O_3_. Al_2_O_3_ will generate easily soluble AlOOH in high-temperature water, which will seriously damage the integrity of the oxide film and cause it to fall off. At the same time, the diffusion of elements during the corrosion process only exacerbates this process. Therefore, the corrosion resistance of the F_0_ alloy in simulated PWR primary water was better than that of the B_0_ alloy.

## Figures and Tables

**Figure 1 materials-15-04975-f001:**
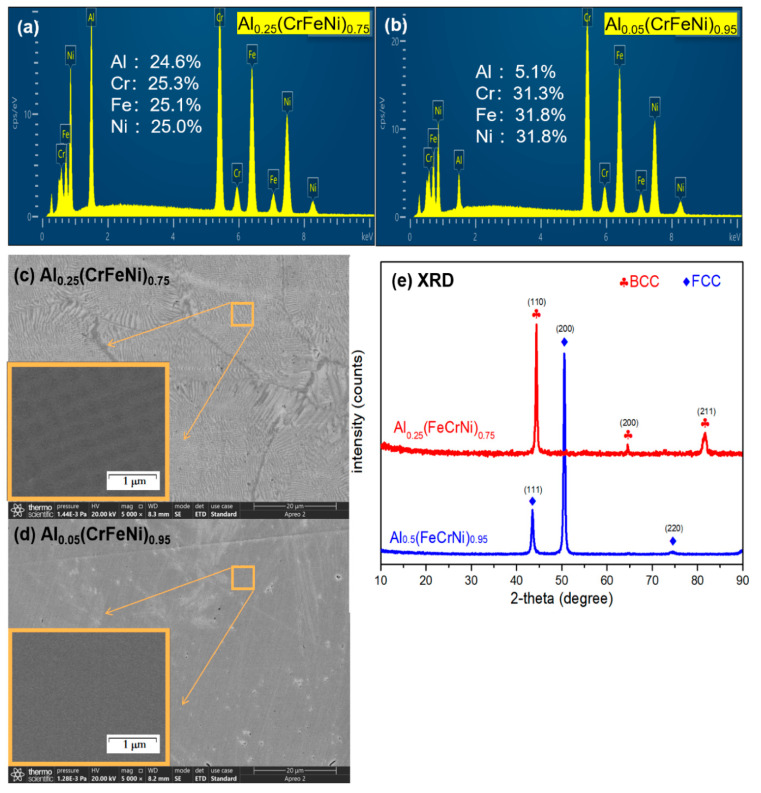
DES of alloys Al_0.25_(FeCrNi)_0.75_ (**a**) and Al_0.05_(CrFeNi)_0.95_ (**b**), surface microstructure (**c**,**d**), and XRD patterns (**e**).

**Figure 2 materials-15-04975-f002:**
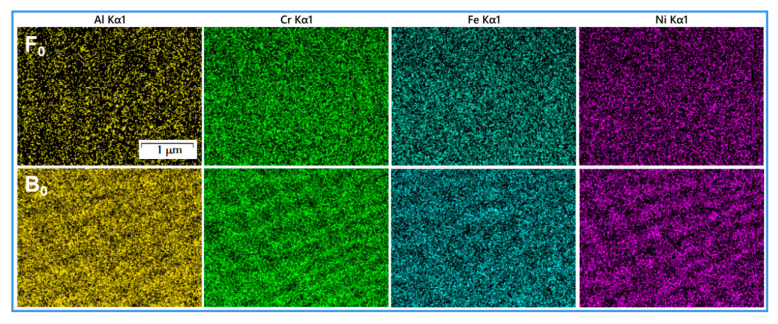
DES mapping of alloys B_0_ and F_0_.

**Figure 3 materials-15-04975-f003:**
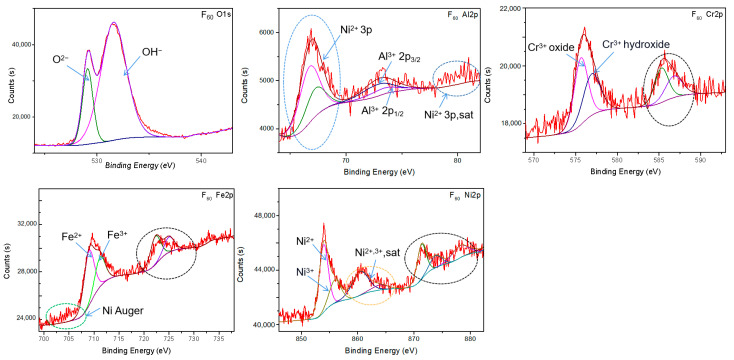
XPS fine spectrum of F_60_ sample (including Al, Cr, Fe, Ni, O).

**Figure 4 materials-15-04975-f004:**
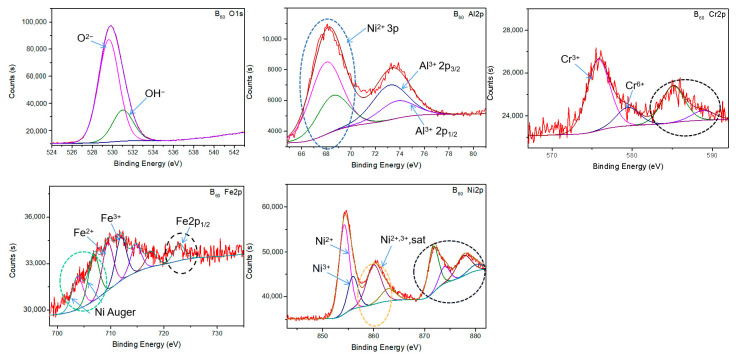
XPS fine spectrum of B_60_ sample (including Al, Cr, Fe, Ni, O).

**Figure 5 materials-15-04975-f005:**
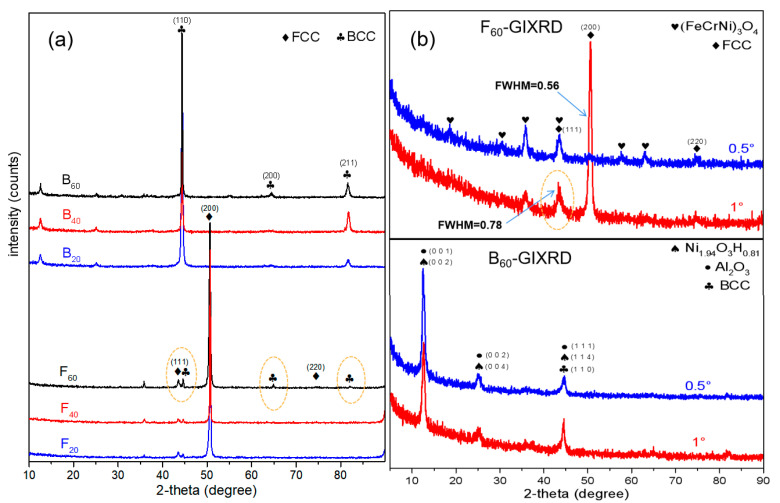
XRD of samples after simulating PWR primary water corrosion (**a**), and GIXRD of F_60_ and B_60_ (**b**).

**Figure 6 materials-15-04975-f006:**
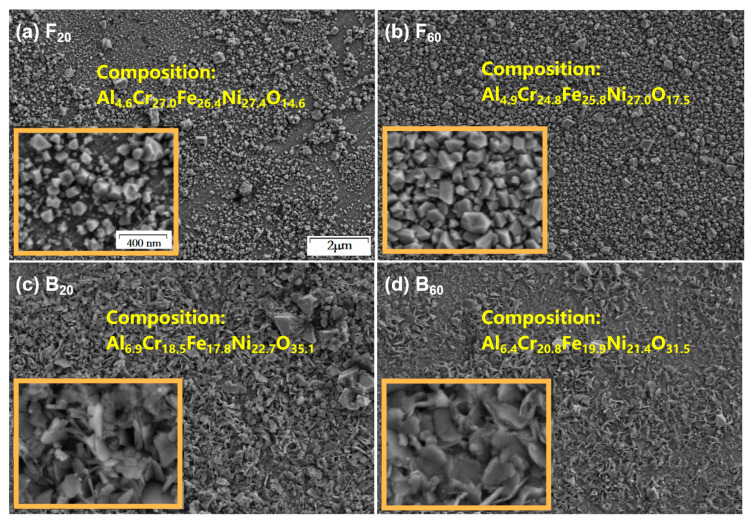
(**a**–**d**) are the surface microstructures (10,000×) of F20, F60, B20, and B60, respectively. The orange box in the lower left corner shows the higher magnification (50,000×). The composition of each sample has been marked in the figure.

**Figure 7 materials-15-04975-f007:**
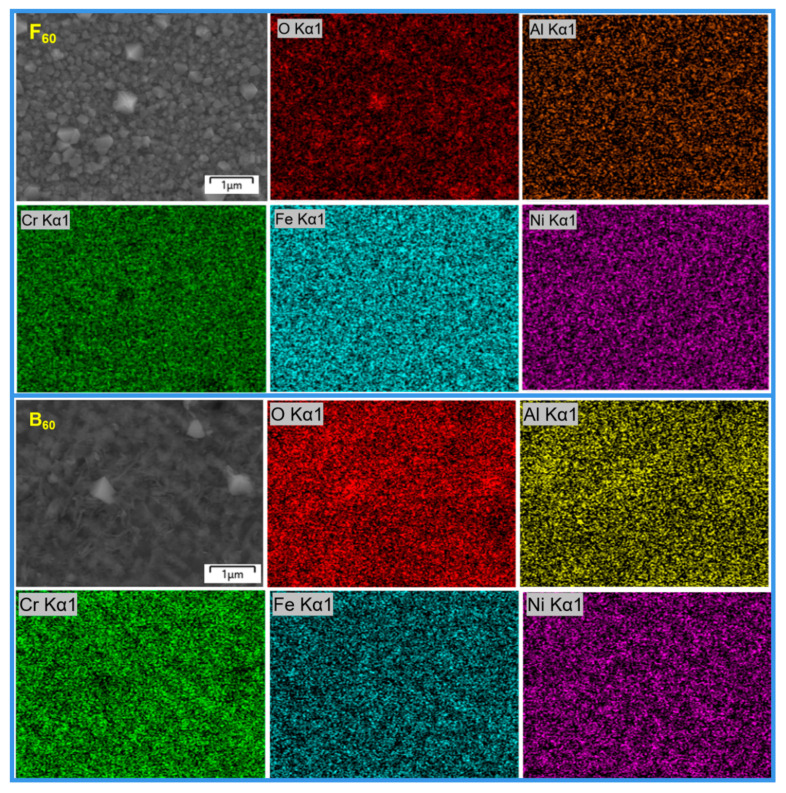
EDS mapping after 60 days of primary water exposure in PWR.

**Figure 8 materials-15-04975-f008:**
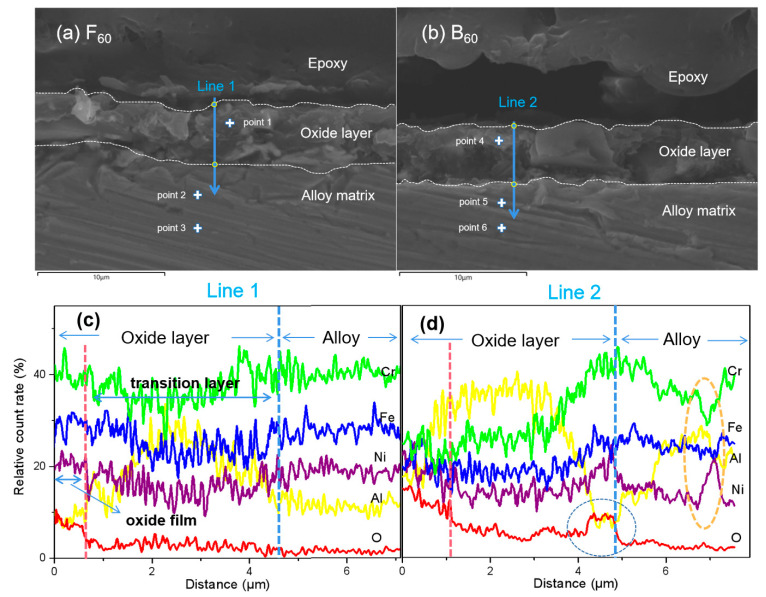
F_60_ and B_60_ cross-sectional microstructure (**a**,**b**). EDS lines of Line 1 and Line 2 (**c**,**d**).

**Figure 9 materials-15-04975-f009:**
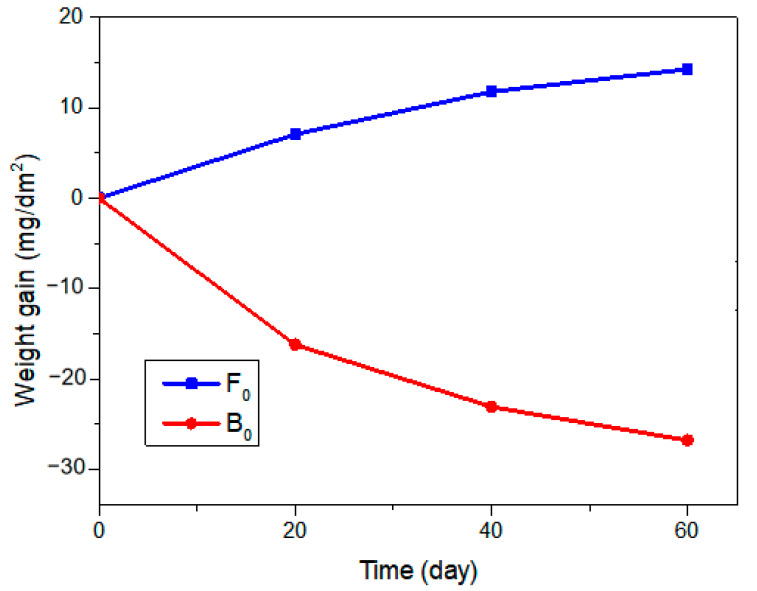
Variation curve of weight gain of F_0_ and B_0_ alloys under PWR primary water with corrosion time.

**Table 1 materials-15-04975-t001:** EDS results for Points 1–6 in Figure 8.

Point	Composition at%				
	Al	Cr	Fe	Ni	O
1	20.88	26.00	25.45	21.93	3.73
2	5.36	30.88	30.98	31.74	1.03
3	5.29	31.39	30.66	31.51	1.16
4	25.54	22.42	22.21	20.80	9.03
5	20.50	29.06	28.29	21.50	0.65
6	23.64	22.94	24.69	27.72	1.01

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
