# Peer review of "Corrosion Behavior of Alx(CrFeNi)1−x HEA under Simulated PWR Primary Water"

_materials, 2022, doi:10.3390/ma15144975_

Round 1
Reviewer 1 Report
The paper investigates the corrosion behavior of Alx(CrFeNi)1-x HEA under simulated PWR primary water. The manuscript has serious flaws and needs a round of major revisions. I will review the revised version afterward:
1- There are some typos and language problems in the manuscript. It should be checked and the language should be checked and revised by a native English speaker who is an expert in the field. In general, the manuscript does not read well and needs extensive revision.
2- About the results of your study on oxidation products, it is needed to cite the relevant papers and explain more about the validity of the interpretation.
3- The error in EDS results was not explained. Also, the reliability of the results should be explained.
4- The other methods for corrosion monitoring should be explained in the introduction such as linear polarization resistance, Tafel extrapolation, .... and explain the comparison between the weight loss technique and those electrochemical techniques. Please use the followings as references: https://doi.org/10.1016/j.rinma.2022.100282
https://doi.org/10.1016/j.apsusc.2006.06.030
5- The sluggish diffusion and segregation during the oxidation process of the HEA should be explained and discussed the effect on the obtained results.
6- Results and discussion are far below the standard of the journal. There is obviously room for more explanation and discussion in most of the results.
Author Response
Dear Editor and Reviewer:
Thanks for your letter and the reviewer’s comments on our manuscript. Those comments are all valuable and very helpful for revising and improving our paper. We have read the comments carefully and made corrections which we hope to meet with approval. Revised portions are marked in red in the new manuscript. The main corrections in the paper and the responds to the reviewer’s comments are appended below.
Responds to the reviewer 1’s comments:
1- There are some typos and language problems in the manuscript. It should be checked and the language should be checked and revised by a native English speaker who is an expert in the field. In general, the manuscript does not read well and needs extensive revision.
Response1: Thanks for reviewer’s comment. The author is not a native English speaker, sorry about that. At present, the article has been compiled in English by professionals, and then carefully corrected by the author. I believe most language problems have been solved. See resubmitted manuscript for details.
2- About the results of your study on oxidation products, it is needed to cite the relevant papers and explain more about the validity of the interpretation.
Response2: Thanks for reviewer’s comment. It is indeed not very rigorous to obtain the phase structure of the oxidized product only by XRD. Therefore, we agree with the reviewer and add the comparative results of similar experiments as a reference. At the same time, we may not have described it clearly. In fact, the determination of the phase structure in this paper is based on a combination of XRD, XPS, and EDS, and is not simply determined by the XRD spectrum. Therefore, the inference of the oxidation products of F60 and B60 should be reasonable.
In this regard, we have partially improved the part of the phase structure analysis in Section 3.1, exchanged the order of the XPS part and the XRD part, and re-described the final conclusion of the oxidation product. Such modifications make the discussion of oxidation products more reasonable. See resubmitted manuscript for details.
3- The error in EDS results was not explained. Also, the reliability of the results should be explained.
Response3: Thanks for reviewer’s comment. Sorry, we may not have fully understood your comment, what is "error in EDS results"? But we did discuss the EDS results in Figure 6 in more detail, making the results more in-depth and the discussion process clearer. The modified part is located in the first and second paragraphs of Section 3.3. See resubmitted manuscript for details.
4- The other methods for corrosion monitoring should be explained in the introduction such as linear polarization resistance, Tafel extrapolation, .... and explain the comparison between the weight loss technique and those electrochemical techniques. Please use the followings as references: https://doi.org/10.1016/j.rinma.2022.100282
https://doi.org/10.1016/j.apsusc.2006.06.030
Response4: Thanks for reviewer’s comment. Electrochemical technology is indeed an effective means in the monitoring of metal corrosion. However, based on the characteristics of the experimental instrument in this study, it is difficult to realize real-time electrochemical monitoring in the autoclave. Therefore, this part of the experiment was not designed during our research. And the focus of this paper is to explore the oxidation products after corrosion, element diffusion and its influence, and the protection of the oxide film to the alloy matrix. Adding an introduction to electrochemical techniques does not improve the rigor of the article. At the same time, it cannot show the advantages of the weight loss technique used in this paper. On the contrary, it may not be conducive to some readers to feel the main content of the article. Therefore, we believe that the electrochemical detection technology is not necessary to focus on the introduction.
5- The sluggish diffusion and segregation during the oxidation process of the HEA should be explained and discussed the effect on the obtained results.
Response5: Thanks for reviewer’s comment. As the reviewer said, the segregation of elements in the alloy has a great influence on the final oxidation product and the overall corrosion resistance of the alloy. Therefore, in the description of Figures 7 and 8, we reconsider the effect of elemental diffusion on the corrosion process. Especially the diffusion of the Al element, because the Al element has the most preferential oxidation order. At the same time, the hydrolysis of its oxidation product Al2O3 will also bring a series of adverse effects on corrosion resistance. The revised part is located in the sixth and seventh paragraphs of Section 3.3. See resubmitted manuscript for details. Such a more in-depth discussion is beneficial to improve the level of the article.
6- Results and discussion are far below the standard of the journal. There is obviously room for more explanation and discussion in most of the results.
Response6: Thanks for reviewer’s comment. Most of the results are discussed in greater depth in the resubmitted manuscript. The focus of the modification includes the discussion of the phase structure of oxides, as well as the discussion of Figure 6, Figure 7, and Figure 8. It is believed that the level of discussion of most of the key results in the article has been greatly improved. See resubmitted manuscript for details.
The Results and Discussion sections of the manuscript have been significantly revised and reviewers are advised to re-read. Due to major changes in the manuscript. Therefore, two versions of the manuscript are provided, namely "Manuscript Draft Version" and "Manuscript Final Version". In the end, the "Manuscript Final Version" shall prevail. "Manuscript Draft Version" is a place for reviewers to track changes.
That’s all of my cover letter, thanks for your reading and we would like to express our great appreciation to you and reviewers for comments on our paper. Looking forward to hearing from you.
Special thanks for your sincere comments.
Best wishes!
First author: Dongwei Luo, Mail: 1510050154@qq.com
Please see the attachment

Reviewer 2 Report
In this manuscript, the authors carried out a detailed corrosion study on Al0.05(CrFeNi)0.95 and Al0.25(CrFeNi)0.75 alloys with various techniques.
The authors need to answer the following questions. After the suggested minor revision below, this work can be considered for publication in “Materials”
11. In Fig.3b, (102), (200) and (220) peaks of B60-GIXRD merge with the background, so the authors can’t claim them as peaks. If the authors want to keep those peaks, better to do GIXRD of those samples with large collection time to increase the intensity of those peaks. Otherwise, simply remove those peaks from the XRD pattern
22. In section 2.3, is “element valence state“ correct statement?
Author Response
Dear Editor and Reviewer:
Thanks for your letter and the reviewer’s comments on our manuscript. Those comments are all valuable and very helpful for revising and improving our paper. We have read the comments carefully and made corrections which we hope to meet with approval. Revised portions are marked in red in the new manuscript. The main corrections in the paper and the responds to the reviewer’s comments are appended below.
Responds to the reviewer 2’s comments:
- In Fig.3b, (102), (200) and (220) peaks of B60-GIXRD merge with the background, so the authors can’t claim them as peaks. If the authors want to keep those peaks, better to do GIXRD of those samples with large collection time to increase the intensity of those peaks. Otherwise, simply remove those peaks from the XRD pattern.
Response1: Thanks for reviewer’s comment. The three peaks (102), (200) and (220) marked in Fig. 3b are comparable in intensity to the curve noise, and are considered to be peaks that are indeed reluctant. Now that the original image has been modified, the label has been removed.
It should be noted that the removal of these three peaks does not adversely affect the discussion and results. The (200) and (220) peaks marked in the original figure are one of the characteristic peaks of BCC. In Figure 3(a), it can be seen that the strong peak in the B0 alloy is (110), and the (200) and (220) signals are relatively weaker. Meanwhile, since a layer of oxide is produced on the surface of B60, these two peaks are hardly visible in the GIXRD of B60, which is reasonable. (102) As one of the characteristic peaks of Al2O3, we know in the following discussion that the Al2O3 produced by the surface oxidation of the B60 sample will react to form easily soluble AlOOH at high temperature. Therefore, there should be very little Al2O3 on the surface. To sum up, these three peaks are very weak, which is in line with the argument before and after the article, so even prolonging the test time of GIXRD may not significantly improve the significance of these three peaks.
However, in order not to cause unnecessary reading troubles to readers, readers may mistakenly think that this place is marked as an actual peak. Agree with the reviewer to delete these three peaks. This will improve the accuracy of the article. The modified part is in the third paragraph of Section 3.2.2. See resubmitted manuscript for details.
- In section 2.3, is “element valence state” correct statement?
Response2: Thanks for reviewer’s comment. As the reviewer said, there is indeed a mischaracterization there. Modifications have been made in the article by reading professional literature in the relevant field. Change the original “element valence state” to “valence of the elements”. The discussion about element valence is mainly in Section 3.2.1 of this paper, so the inappropriate description of the XPS discussion process in Section 3.2.1 has also been modified accordingly. Please reviewer to check.
The Results and Discussion sections of the manuscript have been significantly revised and reviewers are advised to re-read. Due to major changes in the manuscript. Therefore, two versions of the manuscript are provided, namely "Manuscript Draft Version" and "Manuscript Final Version". In the end, the "Manuscript Final Version" shall prevail. "Manuscript Draft Version" is a place for reviewers to track changes.
That’s all of my cover letter, thanks for your reading and we would like to express our great appreciation to you and reviewers for comments on our paper. Looking forward to hearing from you.
Special thanks for your sincere comments.
Best wishes!
First author: Dongwei Luo, Mail: 1510050154@qq.com

Reviewer 3 Report
The authors have investigated "Corrosion behavior of Alx(CrFeNi)1 x HEA under simulated PWR primary water". The Aluminum forming duplex (BCC and FCC) stainless steel (ADSS) is a new material and investigation of its corrosion behavior carries significant importance in nuclear applications. The research paper is good. So, it is recommended for publication with minor revision.
1. The authors should discuss the research work cited in the introduction section.
2. Bulk citations should be avoided, for eg. 7-10, 22-25.
Author Response
Dear Editor and Reviewer:
Thanks for your letter and the reviewer’s comments on our manuscript. Those comments are all valuable and very helpful for revising and improving our paper. We have read the comments carefully and made corrections which we hope to meet with approval. Revised portions are marked in red in the new manuscript. The main corrections in the paper and the responds to the reviewer’s comments are appended below.
Responds to the reviewer 3’s comments:
- The authors should discuss the research work cited in the introduction section.
Response1: Thanks for reviewer’s comment. In this article, our writing logic is as follows.
The introduction part is introduced with the ATF cladding, in which the deficiencies of the traditional ATF cladding are an important turning point, which has been explained in the original text by the method of enumerating one by one. The focus of this work is to use the advantages of HEA to improve the shortcomings of ADSS alloys, so in the second paragraph, the advantages of ADSS alloys are elaborated to highlight the value of choosing it, and at the same time, it shows that its shortcomings highlight the necessity of improving it. On the other hand, the second paragraph presents the advantages of HEA in a general manner.
Therefore, the re-revised manuscript is supplemented with two examples of HEAs being designed as ATF cladding. Increased detail in the introduction. Please refer to the manuscript for detailed changes.
- Bulk citations should be avoided, for eg. 7-10, 22-25.
Response2: Thanks for reviewer’s comment. In response to this comment, after careful research by several authors, it is believed that the quoted part can indeed be trimmed a little. Some highly repetitive and underrepresented citations have been removed, and references have been retyped. The numbers of the deleted citations in the original text are: 5, 8, 17, 20, 22, 23. And according to the actual needs of the article, three references were added again. Changes to the references can be seen at the end of the article. After the citation adjustment, the conciseness and accuracy of article citations have indeed been improved.
The Results and Discussion sections of the manuscript have been significantly revised and reviewers are advised to re-read. Due to major changes in the manuscript. Therefore, two versions of the manuscript are provided, namely "Manuscript Draft Version" and "Manuscript Final Version". In the end, the "Manuscript Final Version" shall prevail. "Manuscript Draft Version" is a place for reviewers to track changes.
That’s all of my cover letter, thanks for your reading and we would like to express our great appreciation to you and reviewers for comments on our paper. Looking forward to hearing from you.
Special thanks for your sincere comments.
Best wishes!
First author: Dongwei Luo, Mail: 1510050154@qq.com

Round 2
Reviewer 1 Report
This version seems better than the previous one. It can be considered for publication after a revision in style and language.